# Research Progress of Mitochondrial Mechanism in NLRP3 Inflammasome Activation and Exercise Regulation of NLRP3 Inflammasome

**DOI:** 10.3390/ijms221910866

**Published:** 2021-10-08

**Authors:** Tan Zhang, Shuzhe Ding, Ru Wang

**Affiliations:** 1School of Kinesiology, Shanghai University of Sport, Shanghai 200438, China; zhangtan@sus.edu.cn; 2Shanghai Frontiers Science Research Base of Exercise and Metabolic Health, Shanghai 200438, China; 3Key Laboratory of Adolescent Health Assessment and Exercise Intervention, Ministry of Education, East China Normal University, Shanghai 200241, China

**Keywords:** exercise, NLRP3 inflammasome, mitochondria, innate immunity

## Abstract

NLRP3 is an important pattern recognition receptor in the innate immune system, and its activation induces a large number of pro-inflammatory cytokines, IL-1β and IL-18 which are involved in the development of various diseases. In recent years, it has been suggested that mitochondria are the platform for NLRP3 inflammasome activation. Additionally, exercise is considered as an important intervention strategy to mediate the innate immune responses. Generally, chronic moderate-intensity endurance training, resistance training and high-intensity interval training inhibit NLRP3 inflammasome activation in response to various pathological factors. In contrast, acute exercise activates NLRP3 inflammasome. However, the mechanisms by which exercise regulates NLRP3 inflammasome activation are largely unclear. Therefore, the mechanism of NLRP3 inflammasome activation is discussed mainly from the perspective of mitochondria in this review. Moreover, the effect and potential mechanism of exercise on NLRP3 inflammasome are explored, hoping to provide new target for relevant research.

The innate immune system is the first defense line against invasion and is essential for maintaining homeostasis. Innate immune cells rely on pattern recognition receptors (PRRs) to recognize pathogen-associated molecular patterns (PAMPs) [1]. The nucleotide-binding oligomerization domain-like receptor protein 3 (NLRP3) is an important intracellular PRRs. Aberrant activation of the NLRP3 inflammasome leads to excessive inflammatory response which is involved in the occurrence and development of a variety of human diseases, such as gout [2,3], osteoarthritis [4], tumor [5,6], type 2 diabetes dillutus (T2DM) [7,8] and neurodegenerative diseases [9,10]. Therefore, it is of great importance to explore the underlying mechanism for NLRP3 inflammasome activation and develop targeted therapeutic strategies based on the molecular mechanism. Current studies showed that various endogenous and exogenous agonists activate NLRP3 inflammasome, while the activation mechanism is elusive, especially, the role of mitochondria in the NLRP3 inflammasome activation is still controversial [11,12]. Additionally, exercise has long been recognized as an efficient intervention to mediate the innate immune response [13]. Generally, chronic low and moderate-intensity exercise benefits the immune system, while high-intensity exercise exhibits the opposite effect. However, there are few studies about the effect of exercise on the NLRP3 inflammasome. Based on this, this paper mainly discusses the mitochondrial mechanism for the NLRP3 inflammasome activation and the effect and potential mechanism of different types of exercise on NLRP3 inflammasome activity are also summarized.

## 1. NLRP3 Inflammasome

### 1.1. Overview of NLRP3 Inflammasome

The NOD-like receptors (NLRs) family is an evolutionarily conserved protein and contains 23 human members and 34 mouse members [14]. NLRs mainly consist of NLRPs and nucleotide-binding and oligomerization domain (NOD/NLRCs) and have typical structural characteristics, such as most NLRs are composed of three domains, namely the N-terminal PYD domain, the middle NBD domain (also known as the NACHT domain, which can hydrolyze ATP to GTP, thus promoting NLR oligomerization) and the C-terminal leucine-rich LRR domain (responsible for ligand recognition) [15] (Figure 1). In 2002, Martinon [16] first proposed the concept of inflammasome. The NLRP3 inflammasome is the most widely studied inflammasome as far and is composed of the NLRP3, apoptosis speck-like protein containing a caspase recruitment domain (ASC) and precursor caspase-1 (precursor caspase-1, pro-caspase-1). ASC is the adapter for the NLRP3 and includes the N-terminal PYD domain and the C-terminal CARD domain [17]. Pro-caspase-1 is the effector for the NLRP3 and consists of the N-terminal CARD domain, the larger catalytic domain in the middle segment (P20) and the smaller catalytic subunit at the C-terminal (P10) [18].The NLRP3 inflammasome is mainly located in the cytoplasm of immune cells such as macrophages, monocytes and T cells, moreover, it is also expressed in some special non-immune cells such as epithelial cells and skeletal muscle cells [18].

### 1.2. The Mechanism for NLRP3 Inflammasome Activation

Under normal physiological conditions, the activity of intracellular the NLRP3 inflammasome is extremely low to maintain a low inflammatory state. The activation of the NLRP3 inflammasome generally includes two steps: “priming” and “activation” [11] (Figure 2). Priming [11]: Once PAMPs or DAMPs are recognized by the corresponding PRRs, nuclear factor kappa-B (NF-κB) translocation is then triggered, which subsequently promotes the transcriptional expression of the NLRP3, IL-1β and IL-18. Moreover, priming of the NLRP3 inflammasome induces post-translational modification of the NLRP3 [19]. Activation [11]: Regarding activation of a variety of endogenous and exogenous agonists, the NLRP3 first undergoes oligomerization and aggregates into the NLRP3 oligomers, then recruits ASC protein through the PYD-PYD domain, subsequently, the fibrotic ASC protein recruits pro-Caspase-1 through the interaction of CARD-CARD domain and induces the self-cleavage of pro-Caspase-1, lastly, activated caspase-1 cleaves pro-IL-1β and pro-IL-18 to produce active inflammatory factors, IL-1β and IL-18. As previously mentioned, the “activation” step of the NLRP3 inflammasome activation can be induced by a variety of endogenous and exogenous agonists, including microbial components such as pathogenic nucleic acid [20], bacterial pore-forming toxins [21] and nigericin [22], crystal and particles such as silica [23] and beta-amyloid [24], endogenous signals such as ATP [25], mitochondrial ROS (mtROS) [26] and cellular ion flux [27]. However, these agonists are structurally and functionally distinct, therefore most researchers believe that these agonists may activate the NLRP3 inflammasome through common signaling pathways. Currently, the well-established mechanisms for the NLRP3 inflammasome activation include intracellular ion flux (such as K^+^ efflux [21], Ca^2+^ flux [28], Na^+^ flux [21] and Cl^−^ efflux [29]), mitochondrial damage [20,26], Golgi disassembly [30], lysosomal disruption [31] and metabolic disorder [11,32]. 

#### 1.2.1. MtDNA and NLRP3 Inflammasome

Mitochondria, the bilayer organelles, are widely distributed in the cytosol of eukaryotes and are the main place for oxidative phosphorylation. Unlike other organelles, mitochondria are regulated by both mitochondrial DNA (mtDNA) and nuclear DNA (nDNA). Each mitochondrion contains multiple copies of mtDNA that are wrapped in proteins called nucleoid complexes [33]. Under normal physiological condition, mtDNA is localized in the mitochondrial matrix. Mitochondrial transcription factor (Tfam) is not only a protein required for nucleoid formation, but also participates in the regulation of mtDNA transcription and replication. In 2011, Nakahira [34] first linked mtDNA to the NLRP3 inflammasome, they found that knockout of microtube-associated protein-1 light chain 3B (LC3B) and moesin-like BCL2-acting Protein 1(Beclin1) led to impaired clearance of damaged mitochondria, thus increasing the release of mtDNA into cytosol and subsequently activating the NLRP3 inflammasome. Moreover, depletion of mtDNA significantly inhibited the NLRP3 inflammasome activation. Further study showed that mtROS generation induced by mitochondrial damage oxidized mtDNA to oxidized-mtDNA (ox-mtDNA), which subsequently released into cytosol where it activated the NLRP3 through binding [20]. Zhong [35] showed that LPS treatment not only induced increased transcription and replication of mtDNA in macrophages, but also led to increased release of ox-mtDNA which subsequently bound with the NLRP3 inflammasome. This may be due to the fact that the new synthesized mtDNA has not been packaged by Tfam to form the nucleoid structure, therefore, it is prone to be oxidized. Taken together, current studies show that mtDNA, as an intracellular DAMPs, releases into cytosol and then activates the NLRP3 inflammasomes, thus triggering immune responses and participating in the development of various inflammatory diseases (Figure 3).

However, how ox-mtDNA releases into cytosol to activate the NLRP3 inflammasome is still elusive. When apoptosis occurs, mtDNA releases from mitochondria into cytosol [36]. Bcl-2-associated K protein (Bak) and Bcl-2-associated X protein (Bax) are the main pro-apoptotic factors of mitochondrial pathway induced apoptosis [37]. Bak is located on the outer membrane of mitochondria and Bax is located in the cytosol under normal physiological conditions. Upon exogenous pro-apoptotic factors, Bax quickly migrates to the mitochondrial outer membrane and then binds with Bak to form the pores, thus changing the permeability of the mitochondrial outer membrane, inducing the release of cytochrome C which triggers cell apoptosis [38]. Notably, recent studies showed that in addition to cytochrome C, mtDNA also released into cytosol through Bak/ Bax pores [39] and the Bak/Bax pores gradually expanded as apoptosis continued which facilitated mtDNA release into cytosol [40]. Additionally, another recent study by Kim [41] showed that the specific mechanisms for mtDNA releasing into cytosol were highly dependent on triggers, since mtDNA binds to voltage-dependent anion selective channel proteins (VDAC) on the mitochondrial outer membrane and then induces VDAC oligomerization to form the pores for mtDNA releasing into cytosol under oxidative stress. 

Taken together, under pathological conditions, mtROS production increases and oxidizes mtDNA to ox-mtDNA. Additionally, mitochondrial proteins form the pores on the mitochondrial outer membrane and allow the release of ox-mtDNA into cytosol where it activates the NLRP3 inflammasome through binding. 

#### 1.2.2. MtROS and NLRP3 Inflammasome

Under normal physiological conditions, electron leakage from the mitochondrial electron transport chain is the main pathway for mtROS generation. By contrast, mitochondria are damaged and results in increased production of mtROS under pathological condition. In 2011, Zhou [42] reported for the first time that mitochondria were involved in the NLRP3 inflammasome activation. Their study showed that autophagy and mitophagy inhibitors blunted the clearance of damaged mitochondria and resulted in increased accumulation of damaged mitochondria, then mtROS production was increased which might activate the NLRP3 inflammasome. Moreover, ATP was also able to activate the NLRP3 inflammasome by promoting the production of mtROS [43]. Fatty acid accumulation induced by a long-term high-fat diet also activated the NLRP3 inflammasome through the AMP-activated protein kinase (AMPK)/autophagy/mtROS pathway [44]. These studies suggest that the increased production of mtROS induced by various factors led to the NLRP3 inflammasome activation. Further research showed that mtROS-induced the NLRP3 inflammasome activation was mediated by Thioredoxin interacting protein (TXNIP) and Thioredoxin (TRX) [26]. TXNIP bound closely with TRX under normal physiological condition, while the high production of mtROS led to TRX oxidation and dissociation from TXNIP; subsequently, the dissociative TXNIP bound to and activated the NLRP3 inflammasome (Figure 3).

It should be noted that some studies found that mtROS was dispensable for the NLRP3 inflammasome activation, as serum β-amyloid induced the NLRP3 inflammasome activation through mtROS-dependent and mtROS-independent pathways [45]. Similarly, some viruses, such as linezolid, influenza virus and encephalomyocarditis virus also activated the NLRP3 inflammasome independently of mtROS [46,47]. Taken together, these results indicate that the role of mtROS in the NLRP3 inflammasome activation is controversial and whether mtROS is required for the NLRP3 inflammasome activation needs to be clarified.

However, although most of the current studies believe that mtROS were generated after mitochondrial injury and ox-mtDNA was released into cytosol, leading to the NLRP3 inflammasome activation. One recent study reported that mitochondria were not involved in the regulation of the NLRP3 inflammasome activation [30]. 

## 2. Effect of Exercise on NLRP3 Inflammasome Activation

Exercise has long been considered an important intervention to regulate innate immune response [13]. Chronic low and moderate-intensity exercise are thought to improve the immune system, while high-intensity exercise impairs it. As an important intracellular PRRs, NLRP3 is an important component of the innate immune system. However, there are few studies on the effect of exercise on the NLRP3 inflammasome activity as far. As shown in Table 1, PubMed was used as the data resource for articles. To identify studies related to the effect of exercise on the NLRP3 inflammasome activity, we used the following keywords: (exercise (title) or training (title) or physical activity (title)) and (NLRP3 (title) or NALP3 (title) or inflammasome (title)). A total of 26 articles were included, and these articles were then classified as endurance training (21), resistance training (2), endurance training combined with resistance training (2), high-intensity interval training (HIIT) (3), acute exercise (2) and exercise preconditioning (1).

### 2.1. Classification of Exercise

Exercise is generally separated into aerobic/endurance and power/strength activities. Endurance training is classically performed against a relatively low load over a long duration [74,75] Resistance training usually includes strength [76] and self-managed loaded exercise [77,78]. Compared with endurance and resistance training, HIIT is new type of exercise and some studies demonstrated that HIIT is more effective to improve blood pressure and aerobic capacity than the traditional chronic moderate-intensity training [79,80,81]. Acute exercise, namely single bout of exercise exhibits different effect depends on the intensity of exercise [82]. It should be noted that the exercise intensity is usually indicated by the ratio of maximum heart rate (max HR) and the percentage of VO_2_max. For example, 40–50% of the max HR indicates low-intensity exercise, 50–70% of the max HR indicates moderate-intensity exercise and >70% of the max HR means high-intensity exercise [83,84]. For VO_2_max, <60% VO_2_max means low-intensity exercise, moderate (60–75% VO_2_max) and high (>90% VO_2_max) [85,86]. Additionally, in line with other studies [87,88], we defined 4 weeks as a timing point for short-time and long-time exercise; short-time exercise means the duration is less than 4 weeks while long-time lasts at least 4 weeks.

### 2.2. Endurance Training and NLRP3 Inflammasome

In 2011, Vandanmagsar [48] first reported that caloric restriction combined with exercise significantly reduced body weight as well as adipocyte volume, improved insulin sensitivity and significantly inhibited the mRNA expression of IL-1β and NLRP3 in subcutaneous tissue of T2DM patients. Moreover, the decrease of mRNA expressions of IL-1β and NLRP3 were positively correlated with the decrease of blood glucose and the improvement of insulin resistance index. These results suggest that moderate-intensity exercise may improve insulin sensitivity by inhibiting the NLRP3 inflammasome overactivation, thus alleviating insulin resistance. Following researchers investigated the effects of different exercises on the NLRP3 inflammasome activity. As shown in Table 1, endurance training was most widely used and treadmill running was the main exercise mode, followed by swimming and voluntary wheel running. It is worth mentioning that moderate-intensity treadmill exercise over 4 weeks was generally demonstrated to significantly inhibit the overactivation of the NLRP3 inflammasome of mice and rats in adipose tissue [49,50], liver [51,52], myocardium [53,54,55] hippocampus [56,57,58,59,60,61], prefrontal cortex [63] and substantia nigra [62] which were caused by metabolic disorder, DEN (diethylnitrosamine) damage, hypoxia, myocardial hypertrophy, aging, myocardial infarction, ovariectomy, Alzheimer’s’ disease, depression, cerebral ischemia and ischemia reperfusion. Similarly, human studies also showed that chronic moderate-intensity running significantly reduced the NLRP3 inflammasome activation in peripheral blood mononuclear cells (PBMCs) of healthy young men, while chronic high-intensity running activated the NLRP3 inflammasome [64]. These results indicate that the effect of endurance exercise on the NLRP3 inflammasome activity mainly depends on the exercise intensity. Chronic moderate-intensity endurance exercise inhibits the NLRP3 inflammasome overactivation which is caused by various pathological factors, while chronic high-intensity endurance exercise induces the NLRP3 inflammasome overactivation, resulting in impaired immune function. Additionally, chronic swimming exercise [65,66] and voluntary wheel exercise [67,68] also inhibited the NLRP3 inflammasome overactivation due to depression, high-fat diet, and chronic kidney disease. 

### 2.3. Resistance Training and NLRP3 Inflammasome

In contrast, ladder climbing is the main form of resistance training, however, there are few studies about resistance exercise and the NLRP3 inflammasome activity compared with endurance training. Two groups compared the effects of chronic endurance training and resistance training on the NLRP3 inflammasome activity. Mardare [49] showed that similar to endurance training, resistance training also inhibited the NLRP3 inflammasome activation in adipose tissue induced by high-fat diet. Moreover, another study showed that chronic resistance training inhibited the NLRP3 inflammasome overactivation in PBMCs caused by aging [69].

### 2.4. Endurance Training Combined with Resistance Training, NLRP3 Inflammasome

Based on these results, Zaidi [70] and Quiroga [71] investigated the combined effect of chronic endurance and resistance training on the NLRP3 inflammasome activation, and the activity of NLRP3 was significantly blunted in PBMCs of obese children despite no effect on adults with T2DM and coronary artery disease. 

### 2.5. HIIT and NLRP3 Inflammasome

Apart from the endurance and resistance training, a growing body of studies showed that HIIT is one of the most effective exercise interventions. Two studies showed that chronic HIIT markedly inhibited the NLRP3 inflammasome overactivation in hippocampus of mice with Alzheimer’s disease [58] and stroke-induced depression [59]. More importantly, HIIT is more effective than endurance training [59] in certain conditions. However, chronic HIIT had no effect on hepatic the NLRP3 inflammasome activation of mice with DEN damage [52] One possibility is that the mice model, detection tissue and exercise duration differs in these studies. 

### 2.6. Acute Exercise and NLRP3 Inflammasome

Furthermore, human studies showed that acute moderate-intensity exhibited no effect on the NLRP3 inflammasome activity of healthy young men, while acute high-intensity exercise activated the NLRP3 inflammasome [64], indicating that the effect of acute exercise on NLRP3 inflammasome activity is affected by a variety of factors and further studies are needed to address this. Additionally, the NLRP3 inflammasome was activated in untrained humans in response to acute exercise, while remained unchanged or decreased in trained humans [72]. 

### 2.7. Exercise Preconditioning and NLRP3 Inflammasome

These above results indicate that moderate exercise benefits the immune system. Additionally, exercise preconditioning were also shown to protect against the NLRP3 inflammasome overactivation induced by acute exercise [73].

Although the effect of different types of exercise on NLRP3 inflammasome activity in various tissues are shown above, these studies focused merely on indicators of NLRP3 inflammasome signaling than the underlying mechanisms.

## 3. Exercise, Mitochondria and NLRP3 Inflammasome

Since most studies suggest that mitochondria are involved in regulating NLRP3 inflammasome activation, it is speculated that exercise adaptation of mitochondria may affect the NLRP3 inflammasome activity. However, there are few reports about the effect of exercise on mitochondria-associated the NLRP3 inflammasome signaling. Previous studies showed that chronic moderate-intensity exercise significantly reduced the expression of inflammatory cytokines tumor necrosis factorα (TNF-α), IL-6, and chemoattractant protein-1 (MCP-1) induced by metabolic disorders, accompanied with increased expression of mitochondrial proteins [89,90,91], while high-intensity exercise resulted in mitochondrial dysfunction and increased secretion of pro-inflammatory factors [92,93]. Additionally, further studies showed that chronic moderate-intensity exercise promoted mitochondrial biogenesis [94,95,96], enhanced antioxidant capacity and inhibited NLRP3 inflammasome overactivation [97,98,99], Moreover, aerobic exercise attenuated cardiac dysfunction by modulating the expression of proteins involved in mitochondrial quality, and NLRP3/caspase-1/IL-1β signaling [53]. These findings suggest that chronic moderate-intensity exercise may reduce mtROS production by regulating mitochondrial quality control (mitochondrial proliferation and activation of mitophagy), improving mitochondrial function, and enhancing the clearance of damaged mitochondria, thereby inhibiting the NLRP3 inflammasome pathway and alleviating excessive inflammatory responses.

## 4. Conclusions and Perspective

The NLRP3 inflammasome is tightly associated with various diseases, therefore thousands of literatures have been demonstrated since first the concept of inflammasome was first proposed in 2002. Regrettably, the exact mechanism for the NLRP3 in Inflammasome activation still remains elusive, Therefore here we summarized the role of mitochondria in the NLRP3 inflammasome activation, hoping to give some direction for NLRP3 inflammasome research These results showed that the NLRP3 inflammasome agonists induced mitochondrial damage which activated NLRP3 inflammasome by producing large amounts of mtROS and releasing ox-mtDNA into cytosol. Moreover, exercise is a well-known method to mediate immune responses. Current studies showed that chronic moderate-intensity endurance exercise, resistance exercise and HIIT significantly inhibited the NLRP3 inflammasome activation induced by various pathological factors, thus improving the innate immunity system. In contrast, acute exhaustive exercise generally activated the NLRP3 inflammasome. Meanwhile, a certain period of exercise preconditioning protected against the NLRP3 inflammasome activation induced by acute exhaustive exercise. However, the mechanisms responsible for the NLRP3 inflammasome activation are still obscure and there are some crucial issues to be solved so far: (1) Although most studies believe that mitochondria are involved in the regulation of the NLRP3 inflammasome, one recent study reported that NLRP3 inflammasome activation was not dependent on mitochondria, therefore the role of mitochondria in NLRP3 inflammasome activation is still controversial. (2) NLRP3 agonists induce mtROS production and ox-mtDNA release into cytosol, thus activating NLRP3 inflammasome, but how these triggers exert such mitochondrial responses is unclear. (3) Most existing studies have shown that exercise regulates the NLRP3 inflammasome activity, but the molecular mechanism responsible for theNLRP3 inflammasome activation is rarely reported. Working out these issues will help to understand the activation mechanism of the NLRP3 inflammasome, thus providing scientific theoretical basis and effective therapeutic targets for the treatment of the NLRP3 inflammasome-related diseases.

## Figures and Tables

**Figure 1 ijms-22-10866-f001:**
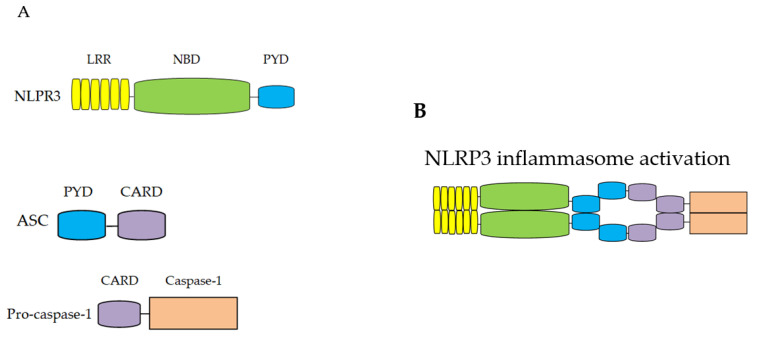
The structure of the NLRP3 inflammasome.

**Figure 2 ijms-22-10866-f002:**
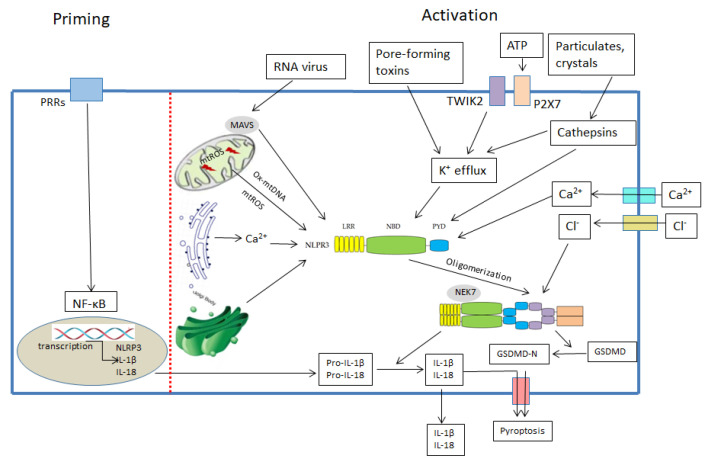
The mechanisms by which NLRP3 inflammasome is activated.

**Figure 3 ijms-22-10866-f003:**
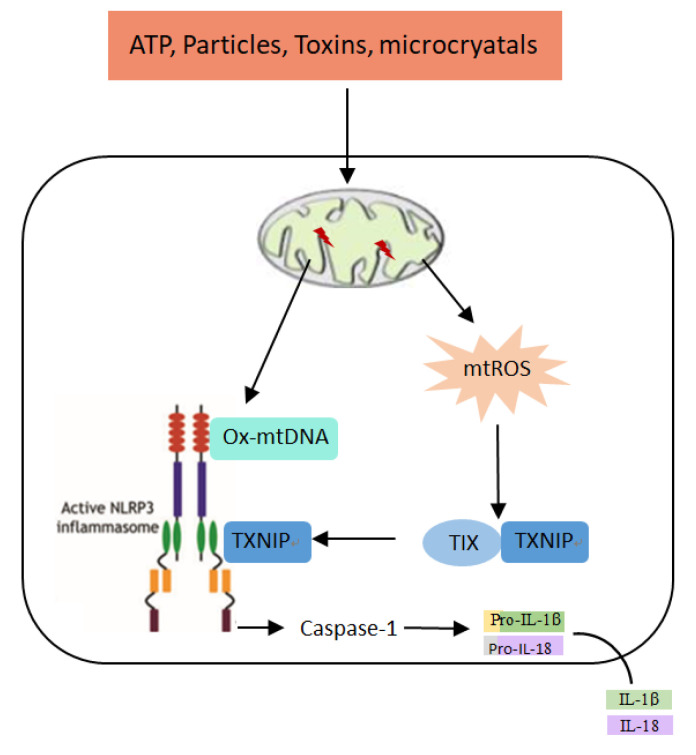
Mitochondria activate the NLRP3 inflammasome through mtDNA and mtROS.

**Table 1 ijms-22-10866-t001:** The effects of exercise on NLRP3 inflammasome activity.

Object	Exercise Type/Intensity/Time	Changes of NLRP3 Inflammasome after Exercise	Reference
Human	Calorie restriction and exercise, 1 year	The mRNA expressions of NLRP3 and IL-1β in subcutaneous fat of T2DM patients were decreased	[48]
**Endurance training**			
Mouse	Treadmill training, 80% VO_2_max, 30 min/d, 5 d/w, 10 weeks.	The mRNA expressions of IL-1β and IL-18 were decreased in adipose tissue of HFD mice	[49]
Mouse	Treadmill training, 70% VO_2_max, 20–50 min/d, 5 d/w, 8 weeks	The protein expressions of NLRP3 in epididymis and subcutaneous fat of HFD nice were decreased	[50]
Mouse	Treadmill training, 12 m/min, 1 h/d, 5 d/w, 12 and 8 weeks	Serum IL-1β, the hepatic mRNA and protein expressions of NLRP3, ASC, caspase-1 and IL-1β were decreased in NASH mice	[51]
Mouse	Treadmill training, 13 m/min, 40 min/d, 5 d/w, 18 weeks	The hepatic mRNA expression of IL-1β was decreased, whereas NLRP3 mRNA expression was increased in mice with DEN damage	[52]
Mouse	Treadmill training, 12–15 m/min, 1 h/d, 5 d/w, 8 weeks	The protein expressions of NLRP3, caspase-1 and IL-1β were decreased in myocardium of mice with myocardial hypertrophy.	[53]
Mouse	Treadmill training, 15 m/min, 1 h/d, 5 d/w, 15–16 weeks	The overactivation of NLRP3 inflammasome signaling was inhibited in myocardium of ApoE−/− mice	[54]
Rat	Treadmill training, from 5 m/min to 10 m/min, 1 h/d, 7 d/w, 12 weeks	Serum IL-1β, the mRNA and protein expressions of NLRP3 and caspase-1 in myocardium of rats were decreased	[55]
Mouse	Treadmill training, 18 m/min, 40 min/d, 5 d/w, 8 weeks	The protein expressions of NLRP3 and IL-1β were decreased in hippocampus of HFD mice	[56]
Mouse	Treadmill training, 15 m/min, 1 h/d, 4 weeks	The mRNA expression of NLRP3 and protein expressions of IL-1β, IL-18, caspase-1 and NLRP3 were decreased in hippocampus of ovariectomized mice	[57]
Mouse	treadmill training, 60% Smax, 5 d/w, 12 weeks	The protein expressions of NLRP3, ASC, IL-1β and caspase-1 were decreased in hippocampus of APP/PS1 mice	[58]
Mouse	Treadmill training, 80% SLT, 7 d/w, 4 weeks	The protein expression of NLRP3 was decreased in hippocampus of mice with depression	[59]
Mouse	Treadmill training, from 6 m/min to 12 m/min, 40 min/d, 5 d/w, 4 weeks	The protein expression of NLRP3 and the ratio of caspase-1/pro-caspase-1 were decreased in hippocampus of mice	[60]
Rat	Treadmill training, from 15 m/min, 30 min/d to 20 m/min, 90 min/d, 6 d/w, 4 weeks	The protein expressions of NLRP3 and IL-1β were decreased in hippocampus of T2DM rats	[61]
Mouse	Treadmill training, 15 m/min, 1 h/d, 5 d/w, 6 weeks	The overactivation of NLRP3 signaling was decreased in substantia nigra of PD mice	[62]
Rat	Treadmill training, from 15 m/min, 30 min/d to 20 m/min, 90 min/d, 5 d/w, 4 weeks	The protein expression of NLRP3 was decreased in prefrontal cortex of T2DM rats	[63]
Healthy young man	Running, Moderate-intensity: High-intensity: 3 d/w, 3 months	The protein expression of NLRP3 and serum IL-1β, IL-18 were decreased after moderate-intensity exercise, while increased after high-intensity exercise	[64]
Rat	Swimming, 1 h/d, 5 d/w, 4 weeks	The protein expression of NLRP3 was decreased in prefrontal cortex of rat with depression	[65]
Mouse	Swimming, 40 min/d, 5 d/w, 12 weeks	The overactivation of NLRP3 signaling induced by HFD was decreased in neuronal tissue of ApoE-/- mice	[66]
Mouse	Voluntary wheel running, 13 weeks	The protein expression of IL-1β in myocardium of HFD mice was decreased	[67]
Mouse	Voluntary wheel running, 6.5 m/min, 1 h/d, 5 d/w, 8 weeks	The protein expressions of NLRP3, ASC, Caspase-1, IL-18 and IL-1β in skeletal muscle of mice with chronic kidney disease were decreased	[68]
**Resistance training**			
Mouse	Ladder climbing, 3 times/d, 1 min for each interval, 5 d/w, 10 weeks	The mRNA expression of NLRP3 in adipose tissue of HFD mice was decreased	[49]
Elderly subjects	Cycle ergometer, 8 weeks	The protein expression of NLRP3 and the ratio of caspase-1/procaspase-1 in PBMCs of elderly subjects were decreased	[69]
**Endurance training combined with resistance training**			
Adults	Endurance training combined with resistance training, 1 year	The activity of NLRP3 inflammasome in serum leucocyte and adipose tissue of adults with T2DM and CAD were unaffected	[70]
Obese Children	Endurance training combined with resistance training, 12 weeks	The protein expressions of NLRP3 and caspase-1 in PBMCs of obese children were decreased	[71]
**High-intensity interval training**			
Mouse	Treadmill training, 85% Smax, 1.5 min; 45% Smax, 2 min, 5 d/w, 12 weeks	The protein expressions of NLRP3, IL-1β and caspase-1 in hippocampus of APP/PS1 mice were decreased	[58]
Mouse	Treadmill training, 60–70% Smax, 5 d/w, 4 weeks	The protein expression of NLRP3 in hippocampus of mice with post-stroke depression was decreased	[59]
Mouse	Treadmill training, 25 m/min, 5 d/w, 18 weeks	The hepatic mRNA expression of IL-1β was unaffected, while NLRP3 mRNA expression was increased in mouse with DEN damage	[52]
**Acute exercise**			
Healthy young men	Running, moderate and high- intensity acute exercise	NLRP3 inflammasome activity in PBMCs of health young men was unaffected after moderate-intensity acute exercise, while increased after high-intensity acute exercise	[64]
Trained/untrained men	A single bout of maximal exercise	The expressions of NLRP3, caspase-1 and IL-1β were increased after acute exercise in untrained men, while decreased or unaffected in trained individuals.	[72]
**Exercise preconditioning**			
Rat	Treadmill training, low, moderate and high-intensity, 8 weeks	The overactivation of NLRP3 inflammasome induced by high-intensity acute exercise was blunted by low, moderate and high-intensity exercise preconditioning, and moderate-intensity exercise exhibits the best effect.	[73]

## Data Availability

Not applicable.

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
