# Peer review of "Research Progress of Mitochondrial Mechanism in NLRP3 Inflammasome Activation and Exercise Regulation of NLRP3 Inflammasome"

_ijms, 2021, doi:10.3390/ijms221910866_

Round 1

Reviewer 1 Report

line 30: for the abbreviation "T2DM", which you also will use later, please write the entire "type 2 Diabetes.."

line 85-86: please, you should enhance the pixel resolution of Figure 2.

line 162: you wrote "studies" but you cited only one.

line 191: please specify "DEN damage"

line 187-196: the sentence is too long.

line 209: you wrote "two studies" but you cited only one.

line 244: please, you have to insert citations for "further studies.."

line 246-274: the same of line 244

line 269 and 272: please, replace the two dot (.) with semicolon (;), before number 2) and 3)

line 255: please, rephrase "...prospect".

you have to insert the citation of your three figures in the text.

Reviewer 2 Report

This manuscript is a review of the understanding of mechanisms of mitochondrial mediated activation of NLRP3 inflammasome as it relates to exercise. The authors describe the various mitochondrial mediated mechanisms and pathways to induce NLRP3 activation. Some concerns are listed below:

  1. Much of the text does not have any references. This needs to be corrected. For example the entire first paragraph of the manuscript has zero references. This occurs throughout the manuscript and are too many to point out one by one. Please add references as appropriate, especially when referring to previous studies.
  2. Please define acronyms at their first mention. For example, T2DM, DAMPs, Tfam, etc.
  3. The table should be condensed. There is too much data listed in the table.
  4. Description of excercise categories mentioned should be included. What exactly is meant by endurance training (how long). How do the authors identify moderate-intensity to high intesity, acute vs. chronic exercise. These descriptions could be added to a small chart.
  5. There should be separate paragraphs for each type of exercise and a review of the literature on the effect of that exercise on NLRP3 activation, mitochondrial ROS and/or DNA, and the pathways that induce NLRP3. 
  6. Significant English editing is needed. There are sentences that are 4 or 5 lines long and grammer/style need improvement.
  7. The conclusion and perspective should pose additional unanswered questions of research interest and potential impact of the current knowledge on health and disease. 

Round 2

Reviewer 2 Report

The authors have adequately addressed all of my previous critiques.